# Young People and the Future: School Students’ Concerns and Hopes for the Future after One Year of COVID-19 in Austria—Findings of a Mixed-Methods Pilot Study

**DOI:** 10.3390/healthcare11162242

**Published:** 2023-08-09

**Authors:** Sonja Fehkührer, Elke Humer, Stefan Kaltschik, Christoph Pieh, Thomas Probst, Gertraud Diestler, Andrea Jesser

**Affiliations:** 1Department for Psychosomatic Medicine and Psychotherapy, University of Continuing Education Krems, 3500 Krems, Austria; sonja.fehkuehrer@edu.donau-uni.ac.at (S.F.); elke.humer@donau-uni.ac.at (E.H.); stefan.kaltschik@outlook.com (S.K.); christoph.pieh@donau-uni.ac.at (C.P.); thomas.probst@donau-uni.ac.at (T.P.); 2Faculty of Psychotherapy Science, Sigmund Freud University Vienna, 1020 Vienna, Austria; 3Österreichischer Arbeitskreis für Gruppentherapie und Gruppendynamik, Fachsektion Integrative Gestalt Therapy Vienna, 1080 Vienna, Austria; g.diestler@gmx.at

**Keywords:** adolescents, COVID-19, concerns about the future, hopes for the future, qualitative research, mixed-methods research

## Abstract

The COVID-19 pandemic resulted in enormous changes in everyday life and numerous burdens for adolescents. This pilot study focused on how young people look to the future in the face of these burdens. Responses to two open-ended questions on concerns and hopes for the future that were part of a larger online survey on the mental health of school students in Austria were analyzed using qualitative content analysis. Subsequently, differences in the experiences of boys and girls, young people with and without a migration background and psychologically stressed and non-stressed young people were examined by applying a mixed-methods approach. Data collection took place from 3 February to 28 February 2021. From a total survey sample of 3052 adolescents aged 14–20, a representative sample according to gender and migration background (*N* = 500) was drawn. Qualitative content analysis revealed several areas of concern about the future, including school-related concerns, concerns about the further development of the pandemic and the associated restrictions, concerns related to a lack of locus of control, health-related concerns and concerns about social relationships. The analysis also indicated young people’s greatest hopes for the future, such as hopes related to further pandemic development, hopes regarding major life goals, school, social relationships and health. Young people’s experiences differ according to gender, migration background and the extent of psychological distress. This study contributes to research on the psychological well-being of adolescents during the pandemic and provides important insights into the subjective experience of young people. It aims to gain a more comprehensive understanding of the concerns and hopes for the future of young people in a time marked by various challenges. The results of the study should be used to develop ideas for measures, such as the expansion of school psychological services and low-threshold support services for students, such as school social work and counseling.

## 1. Introduction

Adolescence is a stage of development characterized by many changes, both physical and psychological. During this time, adolescents are exposed to new and complex social situations, such as peer pressure, romantic relationships and academic stress, which can contribute to psychological distress. All of these factors combined make adolescence a vulnerable stage for mental health [1,2].

The circumstances of the COVID-19 pandemic further contributed to the deterioration in the mental well-being of children and adolescents [1,3,4]. Internationally, a high prevalence of psychological distress could be observed in adolescents during the COVID-19 pandemic, including increased depressive, anxiety, stress and post-traumatic symptomatology [1,2,3,4,5,6,7,8,9,10,11,12,13]. Younger age, female gender and migration background are risk factors for increased psychological distress [1,5,9,13,14,15,16].

Not only did young people’s psychological distress increase significantly during the pandemic, but so did their concerns for the future [17,18]. Concerns and fears were identified in different areas of life.

The fear of contracting the virus, as well as the fear of infecting loved ones, was frequently addressed by children and adolescents surveyed during the pandemic [19,20,21,22,23,24]. Sarkadi et al. [17] found that 60% of the children and adolescents surveyed in Sweden worried that they or their relatives would contract the virus or die as a result. Moreover, 15% of the children and adolescents worried that no cure would be found for the coronavirus. They also expressed fears that the pandemic might be unstoppable and remain for a long time, worsening lives [17,25,26,27]. Similar results were found in other studies [19,28].

Measures and restrictions to contain the coronavirus were accompanied by great uncertainty among children and adolescents. In a study by Hoffmann et al. [29], the young Danish research participants were most concerned about the return to daily life. Moreover, 71% of the German children and adolescents surveyed by Ravens-Sieberer et al. [30] felt burdened by restrictions on contact with friends and family as well as restrictions on leisure activities. This was supported by the results of other studies [18,25,26,29,31,32].

Another area of concern about the future was identified by several studies and addressed the impact of COVID-19 on school performance. Students were concerned about the inability to meet school requirements and having difficulty graduating from high school or transitioning to college [2,18,23,24,25,26,31,33,34,35,36]. Based on a qualitative research design, Jesser et al. [37] found that school-related concerns associated with distance learning were expressed by 63% of Austrian school students surveyed in February 2021. In a German study by Ravens-Sieberer et al. [30], 65% of the surveyed students perceived school and learning as more stressful compared to before the pandemic. In a study by Lehmann et al. [22], 63% of respondents reported learning less during the pandemic.

Concerns about the future were also reported about an uncertain future in the wake of the COVID-19 pandemic [26,28,37]. In a study by Duby et al. [19], female respondents expressed that their dreams and hopes for the future had been shattered. In their view, the pandemic had changed the world. In an Austrian study by Jesser et al. [37], survey participants addressed their fear of lack of prospects, hopelessness and fear of making the wrong decisions. Lehmann et al. [22] showed that 19% of Norwegian adolescents surveyed in mid-May 2020 feared a more difficult future due to the pandemic. This was particularly true for older adolescents (16–19 years). Among other things, the young people were concerned about their future education, career path and career opportunities, and about not being able to achieve their career aspirations in the future. They also addressed challenges in finding a job [25,26,31,37]. Adolescents, particularly those already living in socioeconomically disadvantaged families, were also found to be concerned about their families’ financial circumstances [18,23,25,28,29].

Another concern about the future raised by the adolescent participants in Sarkadi et al.’s study [17] was the feared impact of the COVID-19 pandemic on democracy. It was also found that young people’s trust in politicians was decreasing due to their management decisions during the pandemic.

Several studies have found differences in COVID-19-related concerns about the future between boys and girls and between young people with and without a migration background. Young women and adolescents with migration backgrounds were generally more likely to be concerned than male students [26,28,36,38,39].

Most studies looking at the impact of the COVID-19 pandemic on young people’s concerns and hopes for the future have used quantitative research designs. Few studies have focused on the subjective accounts of adolescents in the wake of the pandemic. This pilot study aimed to gain a deeper insight and understanding of Austrian school students’ concerns and hopes for the future. Open-ended questions were used to encourage free narratives from young people and to get a picture of the range of subjective concerns and hopes for the future that young people are experiencing as the pandemic progresses. The online survey format and sampling design aimed to ensure that we reached many diverse young people, giving a representative picture of their concerns and hopes for the future. Differences in concerns and hopes for the future according to gender, migration background and level of psychological distress were also explored.

This pilot study contributes findings from Austria to the existing knowledge base. The open-ended qualitative approach allowed for a broad view of self-reported concerns and hopes for the future, while the subsequent mixed-methods analysis complemented the findings by looking at differences between genders, young people with and without a migration background, and mentally distressed and non-distressed young people. The findings and lessons learned from our pilot study are intended to form the basis for further in-depth qualitative research in the future.

## 2. Materials and Methods

### 2.1. Research Design

We conducted a cross-sectional online survey between 3 February and 28 February 2021, among 14- to 20-year-old students. The study was part of a larger study in which the current psychological state of adolescent students, as well as their concerns and hopes for the future, were surveyed. We obtained approval from the ethics committee for this study (protocol code EK GZ 41/2018–2021).

The study was carried out using Research Electronic Data Capture (REDCap) [40,41]. and comprised 67 items. While most items were to be answered using rating scales, four open-ended questions provided school students with options to express their thoughts in their own words. Questions addressed respondents’ current concerns, their concerns and hopes for the future, and helpful coping strategies. Quantitative results have already been published [42,43], as well as the results of the two open-ended questions related to current concerns and helpful coping strategies [37].

The study was conducted as an exploratory sequential mixed-methods design. To shed more light on young people’s concerns and hopes for the future, we investigated the following two open-ended questions: (1) What is your greatest concern when you think about the future? (2) What do you hope for most when you think about the future? The results of the qualitative analysis informed the quantitative analysis, which explored differences in the categories of concerns and hopes for the future according to gender, migration background and level of psychological distress.

### 2.2. COVID-19-Related Situation in Austrian Schools during the Period of Study Implementation

When the survey was launched on 3 February 2021, a third lockdown was in place in Austria [44]. As in the previous lockdowns, people were no longer allowed to enter public spaces except to cover their basic needs. All non-essential shops as well as all facilities such as cafés, restaurants, public baths and other sports facilities were closed, and cultural and sports events were canceled. There were strict contact restrictions and curfews that largely brought public life to a standstill. Schools and universities were closed and switched to distance learning. Mouth-nose protection had to be worn in public places, shops and on public transport [45]. After the end of the lockdown on 7 February 2021, schools were reopened in a shift system to minimize the number of students in attendance [46]. The school-leaving examination, the final examination in the Austrian education system, taken by students at the end of their secondary education, was postponed by three weeks to give them more time to prepare. Due to the difference in conditions between distance learning and classroom teaching, the examination content was decreased by 33% [47].

### 2.3. Selection of the Sample

The Department of Psychosomatic Medicine and Psychotherapy at the University of Continuing Education Krems conducted the online study with support from the Federal Ministry of Education, Science, and Research. Secondary school students were made aware of the online survey through social media channels maintained by the university. Participation was voluntary and without incentives. Participants were asked to agree to the privacy policy as well as to confirm their age of 14 years or older.

The total sample of *N* = 3052 was representative of Austria by region, but not by gender and migration background. The average age of the participants was M = 16.47 (SD = 1.44) years. Here, 70.1% of the sample were female, 28.1% male and 1.8% non-binary. Furthermore, 16.6% of them had a migration background. Based on data from Statistics Austria [48] a representative sample of 500 adolescents was stratified and proportionally weighted by gender and migration background, as numerous studies have demonstrated that adolescent mental health is significantly moderated by these two variables [1,5,9,15,16]. Furthermore, only participants who answered both questions were selected.

### 2.4. Data Analysis

#### 2.4.1. Qualitative Data Analysis

The data were subjected to a conventional qualitative content analysis approach [49] and analyzed within an iterative framework for qualitative data analysis [50]. To increase transparency [51] and to facilitate discussion and revision of categories within the research team, we used the software Atlas.ti 23 [52]. In the first step, all responses in the sample were read carefully by one coder (SF). Individual responses to open-ended questions ranged from single words to entire paragraphs. In the process of reading, she composed the first list of inductive categories. In the next step, these categories were sorted thematically and assigned to main categories at a higher level of abstraction. The resulting preliminary category system was iteratively discussed and revised by the research team. Subsequently, SF coded the entire data material using the category system. In some cases, additional inductive categories were added. Text passages that could not be clearly assigned were discussed with the research team. After coding all the data, the category system was reviewed by the research team. The focus was on checking categories for their distinctiveness, evaluating the assignment of subcategories to higher-level main categories and merging smaller categories into more abstract categories.

Finally, the analysis resulted in 11 main categories and 40 subcategories for concerns about the future and 11 main categories and 42 subcategories for hopes for the future.

#### 2.4.2. Quantitative and Mixed-Methods Analysis

Following qualitative analysis, we examined differences based on gender, migration background and the degree of psychological distress using a mixed-methods approach. Mixed-methods analysis was carried out with SPSS version 26 (IBM Corp, Armonk, NY, USA). Chi-squared tests (χ^2^) or Fisher’s exact tests were used to examine:Differences in the proportion of female and male school students reporting in main or subcategories of concerns about the future and hopes for the future.Differences in the proportion of school students with or without migration background in reports in main or subcategories of concerns about the future and hopes for the future.Differences in the proportion of school students with or without psychological distress in reports in main or subcategories of concerns about the future and hopes for the future.

The significance level for all tests was set at 5%. All tests were performed two-tailed. For the analyses of differences in students with or without psychological distress students were categorized into those exceeding the cut-offs for clinically relevant symptoms of depression, anxiety, or insomnia vs. those scoring below the cut-offs for clinically relevant symptoms of depression, anxiety and insomnia. Detailed information on the assessment of symptoms of depression, anxiety and insomnia and the cut-offs applied is provided in our companion paper [42].

## 3. Results

### 3.1. Sample Description

The sample of 500 participants consists of 50% (*n* = 250) female (compared to 49% in the general population of 15–29 years old females in Austria) and 50% male school students. Of the participants, 30.8% (*n* = 154) had a migration background (compared to 28% of the general population aged 15–29 in Austria). The average age of the participants was M = 16.50 (SD = 1.50) years. Further sample characteristics are summarized in Appendix A.

### 3.2. Qualitative Results

#### 3.2.1. Concerns about the Future

In the following, we report the results related to the first question: “What is your greatest concern when you think about the future?” The results are presented in Figure 1 and Appendix A and are now described in more detail.

##### School-Related Concerns about the Future

With *n* 135 (27.0%) mentions, school-related concerns about the future are the largest category. It consists of four subcategories.

Most respondents, *n* = 66 (13.2%), expressed their fear of not being able to graduate from school or having difficulties with the school-leaving examination. *N* = 55 (11.0%) expressed concerns about their future academic performance. They worried about not achieving learning objectives due to COVID-19 measures, such as distance learning. *N* = 13 (2.6%) mentioned concerns about the future regarding school in general, without elaborating on what these concerns related to. *N* = 8 (1.6%) expressed concerns related to the future organization of the school. They were worried about the continuation of distance learning, of being taught in alternating groups (shift system) and about the school closing again. On the other hand, respondents were also concerned about the reopening of schools.

##### Pandemic-Related Concerns about the Future

*N* = 95 (19.0%) respondents mentioned concerns about the future which were related to the COVID-19 pandemic. The main category comprises four subcategories.

The largest subcategory, addressed by *n* = 43 (8.6%), includes concerns related to the further development of the pandemic. Respondents mainly expressed concerns that the situation might worsen or that the pandemic might never end. Further *n* = 32 (6.4%) respondents uttered concerns about pandemic-related restrictions, e.g., the continuation of the restriction to enter public spaces, contact restrictions, limited opportunities to travel and the burden of wearing masks. *N* = 26 (5.2%) expressed concern about whether it will ever be possible to return to normal life as it was before the pandemic. Finally, *n* = 8 (1.6%) raised the issue of vaccination, expressing both concerns that not enough people would be vaccinated and fear that possible compulsory vaccination would come with greater restrictions for the unvaccinated.

##### Concerns about a Lack of Locus of Control in the Future

*N* = 81 (16.2%) respondents expressed concern about having no or insufficient power to shape their own lives and pursue their goals in the future. We have subsumed five subcategories under the more abstract main category “lack of locus of control”.

*N* = 56 (11.2%) respondents referred to their fears of failure, fears of not achieving their goals, of making the wrong decisions, or of disappointing themselves and their families. *N* = 14 (2.8%) expressed uncertainty about their future. *N* = 9 (1.8%) showed a pessimistic attitude toward the future in the form of hopelessness, resignation or negative thinking about the future. *N* = 8 (1.6%) stated that they had no life plan of their own. *N* = 2 (0.4%) were worried about not being able to live independently in the future.

##### Work-Related Concerns about the Future

*N* = 81 (16.2%) respondents expressed concerns regarding their future work situation. We subsumed four subcategories.

*N* = 34 (6.8%) were concerned about not being able to find a job that they like and that offers good working conditions. Respondents indicated that they were concerned about not being able to pursue their “dream job”. *N* = 28 (5.6%) expressed concern about not being able to find a job at all, partly due to the more difficult conditions caused by the pandemic. *N* = 16 (3.2%) reported uncertainty about their future career after leaving school. *N* = 7 (1.4%) mentioned concerns about the future in relation to work, without going into detail.

##### Concerns about the Future Related to Physical and Mental Health

Another category, mentioned by *n* = 64 (12.8%), summarizes concerns related to physical and mental health in the future. Responses were assigned to four subcategories.

Several respondents, (*n* = 31, 6.2%), described concerns about their mental health, including fears that mental health problems might resurface or worsen. Respondents also expressed concern that missing out on adolescence due to the pandemic would have a negative long-term impact on their mental health. Further *n* = 18 (3.6%) raised concerns about being unhappy and joyless in the future. *N* = 13 (2.6%) were concerned about becoming physically ill or even dying. *N* = 7 (1.4%) worried about the mental or physical well-being of others.

##### Concerns about the Future Related to Social Relationships

*N* = 47 (9.4%) respondents expressed concerns about their social relationships in the future.

The subcategories illustrate that *n* = 37 (7.4%) were afraid of being alone, of not making friends or of losing important family members or friends. *N* = 7 (1.4%) expressed concerns about conflicts in their family of origin. *N* = 6 (1.2%) were worried about their relationship—that it would end or that they would not find a partner or the right partner. *N* = 3 (0.6%) were worried that they would not be able to start a family in the future.

##### Concerns about the Future Regarding Further Education

Concerns about the future in relation to further education were named by *n* = 44 (8.8%). Four subcategories were defined.

*N* = 17 (3.4%) expressed uncertainty about their further education after leaving school, e.g., whether or not to study or which subject to study. *N* = 11 (2.2%) were worried about their academic performance at university. Respondents were concerned about having educational deficits caused by the pandemic, which could affect their academic success. *N* = 10 (2.0%) reported concerns regarding their chances of being accepted into a university or not being able to study abroad due to travel restrictions. *N* = 6 (1.2%) mentioned concerns about the future in relation to their studies, without going into more detail.

##### Concerns about the Future Related to Social and Political Developments

Another category mentioned by *n* = 39 (7.8%) respondents, summarizes concerns about future social and political developments. Respondents were assigned to four subcategories.

*N* = 16 (3.2%) were concerned about climate change. *N* = 14 (2.8%) referred to the political situation. *N* = 14 (2.8%) were concerned about the further development of the current economic and financial crisis and *n* = 4 (0.8%) expressed concerns about possible further pandemics and the occurrence of other disasters.

##### Further Concerns about the Future

*N* = 34 (6.8%) addressed their financial situation as a concern about the future. *N* = 11 (2.2%) expressed other concerns, e.g., concerns about compulsory military service or about the coming out. *N* = 23 (4.6%) stated that they were not worried about the future or that they did not think about the future or could not say anything about it (“I don’t know”).

#### 3.2.2. Hopes for the Future

In the following section, we report the results of the second open-ended question: “What do you hope for most when you think about the future?” The results are illustrated in Figure 2 and Appendix A. They are described in detail below.

##### Pandemic-Related Hopes for the Future

Most hopes for the future (*n* = 181, 36.2%) are directly related to the COVID-19 pandemic. Within the main category, five subcategories were formed.

The largest subcategory, with *n* = 92 (18.4%) mentions, relates to the desire to return to normality and life as it was before the pandemic.

*N* = 69 (13.8%) related their hopes to the end of the measures. *N* = 55 (11.0%) wanted the COVID-19 pandemic to end. They wished for the pandemic to be brought under control or for the coronavirus to be defeated. *N* = 7 (1.4%) wanted a higher vaccination rate. *N* = 4 (0.8%) wanted reassurance—more knowledge about what will happen next or a fixed date when the lockdown will end.

##### Hopes for the Future Regarding Major Life Goals

The second largest category of hopes for the future, with *n* = 142 (28.4%) mentions, relates to various general life goals young people want to achieve. Five subcategories were defined.

Most respondents (*n* = 83, 16.6%) wanted to be satisfied with their lives. They hoped for a good life, using adjectives such as “happy” (respondent 3046), “cheerful” (respondent 2316), “lighthearted” (respondent 968) and “satisfying” (respondent 1837). *N* = 40 (8.0%) expressed a desire for self-actualization. Respondents wished for success. Others wanted to make the right choices about their lives, realize their dreams and achieve their goals for the future. *N* = 14 (2.8%) wanted autonomy, i.e., an independent, self-determined way of life. Respondents expressed this as follows: “to be self-reliant” (respondent 826), “to be able to lead a self-determined life” (respondent 1639) or “to be independent of others” (respondent 2714). *N* = 11 (2.2%) reported hopes for the future that were subsumed under the category “personal strength”. The following hopes were expressed: “courage” (respondent 2389), “hope” (respondent 2389), “confidence” (respondent 3052) and “self-confidence and strength” (respondent 1098). *N* = 5 (1.0%) expressed a desire for a life full of experiences and a lot of impressions.

##### Hopes for the Future Relating to Social Relationships

Another category, comprising four subcategories, summarizes hopes for the future relating to social relationships (*n* = 95, 19.0%).

*N* = 52 (10.4%) expressed the wish to have more contact with their friends in the future, to continue to have good friends or to make new friends. *N* = 30 (6.0%) expressed the wish to start their own family or to live in a happy family with their children. *N* = 17 (3.4%) wished to be loved, to find or keep a partner, to have a fulfilling relationship or to marry their current partner. *N* = 12 (2.4%) wished for more contact with their family of origin or for greater family cohesion or the resolution of family conflicts.

##### Health-Related Hopes for the Future

*N* = 89 (17.8%) respondents expressed hopes for the future related to health, which were classified into five subcategories.

*N* = 34 (6.8%) expressed a desire for well-being, e.g., rest, relaxation, less stress and freedom of worry. *N* = 32 (6.4%) wished for physical health or a long life. *N* = 16 (3.2%) wished for the health and well-being of others, especially their loved ones and friends. A further *n* = 11 (2.2%) expressed a desire to improve their fitness or lose weight in the future. *N* = 6 (1.2%) wanted to maintain or regain their mental health.

##### School-Related Hopes for the Future

*N* = 71 (14.2%) respondents expressed hopes for the future regarding their school situation. Four subcategories were formed.

Several respondents (*n* = 34, 6.8%) wanted to complete their school-leaving examinations or graduate successfully from secondary school. *N* = 18 (3.6%) referred to their future school performance, expressing a desire to get good or better grades, to pass exams and thus pass the current school year. *N* = 18 (3.6%) commented on school organization and wished for a return to normal school attendance. Only a few expressed the opposite wish, that schools would remain closed and that there would be no exams, or that school would continue to be run in shifts *n* = 8 (1.6%) expressed the wish to have less stress at school and more free time in the future.

##### Hopes for the Future Related to Lifestyle

Another category summarizes hopes for the future related to lifestyle (*n* = 70, 14.0%). We differentiated five subcategories.

Several respondents within this main category (*n* = 38, 7.6%) wanted to be financially secure and to have no financial problems *n* = 17 (3.4%) wanted to travel, see the world, move abroad to live, study or work, or return to their home country in the future. *N* = 9 (1.8%) referred to their future living situation, mainly expressing a desire to live in a house in the future. *N* = 6 (1.2%) wished for a secure future or stability, without elaborating on what this meant to them. *N* = 6 (1.2%) expressed a desire to get a driving license or own a car.

##### Work-Related Hopes for the Future

*N* = 54 (10.8%) expressed hopes regarding their future work situation.

Several respondents (*n* = 43, 8.6%) wanted a good job that they enjoyed, with good working conditions and a good salary. *N* = 6 (1.2%) wanted to find a job at all and *n* = 5 (1.0%) wanted to be professionally successful and have a career.

##### Hopes for the Future Regarding Further Education

*N* = 34 (6.8%) respondents stated hopes for the future regarding their further education.

*N* = 17 (3.4%) expressed general hopes regarding their further education, such as getting a place at university. *N* = 12 (2.4%) expressed hopes about their future careers, such as finding a suitable career for themselves. *N* = 5 (1.0%) expressed the wish that they would do well in their studies.

##### Hopes for the Future Related to Social and Political Developments

Another category including five subcategories describes hopes related to future social and political developments (*n* = 29, 5.8%).

Concerning social developments, *n* = 16 (3.2%) expressed hopes for global cooperation, social cohesion or world peace. *N* = 6 (1.2%) referred to political development, wishing for better government, communism or less right-thinking people. *N* = 6 (1.2%) referred to climate change and expressed the hope that people would recognize the urgency of the issue and act. *N* = 2 (0.4%) wished for economic growth. Further *n* = 2 (0.4%) wished for an end to new pandemics or other crises.

##### Other Hopes for the Future

*N* = 11 (2.2%) expressed other, sometimes less differentiated wishes, such as that many or all things should change. Two respondents said that they would like to belong to the opposite sex in the future without suffering negative consequences. *N* = 9 (1.8%) stated that they had no wishes for the future or were unable to express any wishes for the future and answered, “I don’t know”.

### 3.3. Quantitative and Mixed-Methods Results

#### 3.3.1. Differences According to Gender

The calculated Chi-squared tests, depicted in Appendix A, showed significant gender differences related to concerns and hopes for the future in both main and subcategories. Female respondents predominantly described themselves as more concerned about the future than male respondents. The pandemic-related concern about whether it will ever be possible to return to a normal life, as well as concerns with regard to their chances of accessing university, were mentioned more frequently by female respondents. Lack of locus of control (main category) and fear of failure, physical and mental health concerns (main category), and the associated subcategories of concerns regarding mental health and becoming unhappy in the future were also mentioned more often by female respondents.

In comparison, male respondents stated more frequently concerns about social and political developments (main category) or not having any concerns, compared to female respondents.

#### 3.3.2. Differences According to Migration Background

The statistically significant differences in the main and subcategories according to migration background in relation to mentioned concerns and hopes for the future are presented in Appendix A.

In comparison to the other two characteristics, gender and degree of psychological distress, fewer statistically significant differences were found. Respondents with a migration background described concerns related to a lack of locus of control (main category) and fear of failure to a greater extent, compared to respondents without a migration background.

On the other hand, respondents without a migration background described a higher degree of pandemic-related concerns about the future (main category) and pandemic-related hopes for the future (main category), particularly the desire for an end to pandemic-related measures. Concerns about the future related to social and political developments were also more often reported among respondents without a migration background. Respondents with a migration background expressed a higher degree of hopes for the health of others, especially their family and friends.

#### 3.3.3. Differences According to the Degree of Psychological Distress

The statistically significant differences in the main and subcategories according to the degree of psychological distress are presented in Appendix A.

In our study, a correlation was found between a higher degree of psychological distress and concerns about the future related to school (main category), especially the concern of not passing the school-leaving examination. Health-related concerns about the future (main category), especially fear of future psychological distress, were also more often expressed by psychologically distressed respondents. Additionally, fear of failure was associated with higher psychological distress. Psychologically distressed adolescents expressed major life goals (main category) more frequently and wished for life satisfaction.

The group of psychologically unstressed respondents expressed concerns about the future related to the vaccination for protection against COVID-19 more often than distressed respondents. They also expressed hopes about future social developments more frequently. This group also more often reported not being concerned about the future.

## 4. Discussion

The present study explored school students’ concerns and hopes for the future after the first year of the pandemic. In addition, the study investigated whether there were differences regarding expressed concerns about and hopes for the future based on gender, migration background and mental health, as often found in the literature [1,5,9,13,14,15,16].

### 4.1. School-Related Concerns and Hopes for the Future

We found that the most common area of concern regarding the future was related to school (27%). The findings illustrate how much young people’s mental well-being depends on academic performance and the learning environment, with individual differences also evident.

Our findings are comparable to other studies. Internationally, adolescents expressed significant concerns about the impact of the pandemic on school attendance and academic performance [22,25,31,32,35,36,37,53]. Compared to our study, Riiser et al. [54] found in a qualitative study with Norwegian adolescents aged 16–18, that the constant change between different forms of teaching and the lack of continuity harmed students’ work habits, motivation, learning progress and learning outcomes. School-related worries were also identified as a risk factor and a strong predictor of psychological distress among adolescents [53,55]. However, other studies have also shown that some students experienced positive effects from the pandemic-related changes in the education system. While 46% of students surveyed in the Swiss Corona Stress Study reported feeling burdened by distance learning, 41% of respondents reported experiencing relief as a result [53]. In addition, the complex demands of personal life were simplified by measures such as contact restrictions, which make it easier for adolescents to concentrate on schoolwork [56].

School-related hopes for the future were mentioned less frequently (14%) than would be expected given the prevalence of school-related concerns. It may be that school students associated their worries about school with the pandemic and its restrictions. This would explain the frequency of pandemic-related hopes for the future (36%). As with concerns about the future, some adolescents expressed a wish for schools to remain closed in the future. Other studies with children and adolescents also indicated improved well-being for some of them during lockdowns [57,58]. Students reported observing progress in self-organization, independent learning and computer skills [59].

### 4.2. Pandemic-Related Concerns and Hopes for the Future

Another relevant aspect of our study relates to school students’ concerns about the future regarding the COVID-19 pandemic. Concerns mainly revolved around the further development of the pandemic and that the pandemic-related restrictions might continue.

Similar results were found by Magson et al. [60]. Young people were more concerned about government restrictions to contain the virus than about the virus itself. Concerns about when life will return to normal are also reflected in another study from Austria [27] and in international comparisons [19,29].

### 4.3. Concerns about the Lack of Locus of Control and Hopes Related to Major Life Goals

Another central theme of our study is young people’s sense of locus of control. Internal locus of control is defined as the belief that the outcome of life events depends on one’s actions and external locus of control is the belief that chance and powerful others control one’s life [61]. Research findings during the COVID-19 pandemic in adults suggest that an internal locus of control contributed to a reduction in pandemic-related psychological distress, whereas an external locus of control increased pandemic-related psychological distress [29,61,62,63]. In their study of adults, Würtzen et al. [64] showed that internal locus of control decreased during the pandemic, especially in younger adults (18–24 years).

In our study, young people also described a lack of internal locus of control and, in particular, a fear of failure. Their concerns can be linked to hopes for the future that relate to young people’s major life goals, such as the desire for life satisfaction, self-actualization and autonomy. The high frequency of these categories should be seen in the context of decreased life satisfaction among young people during the pandemic [60,65,66,67]. Fear of failure has been found to be associated with lower life satisfaction [68,69]. On the other hand, students who felt more competent and supported in their autonomy reported higher well-being [59].

### 4.4. Physical and Mental Health-Related Concerns and Hopes for the Future

An issue that has already received considerable attention in previous research is the concern about contracting COVID-19 in the future or the possibility of close family members contracting the disease. Both aspects were mentioned by only a few adolescents in our study. Other studies have also shown that young people have little fear of contracting COVID-19 themselves [22,24,25,29,31], which may be explained by the lower burden of COVID-19 symptoms in children and adolescents [70]. 

In our study, health-related concerns about the future focused mainly on the concern that one’s mental well-being might (again) deteriorate. Young people’s health-related hopes for the future were also mainly related to their psychological well-being. This is consistent with research showing a significant impact of the pandemic on young people’s mental well-being and an increase in mental health concerns [17,25,42,54,71,72,73].

### 4.5. Concerns and Hopes Related to Social Relationships

Another aspect of young people’s lives that has been strongly affected by COVID-19 measures is their friendships [25,27,29,60,74]. In our study, young people expressed concerns about the future regarding their social relationships. Being with friends is significantly related to adolescents’ well-being and quality of life [59,75,76,77,78]. Spatial distance and social deprivation can have profound psychological effects [60,79,80,81,82]. While it is plausible to assume that adolescents’ extensive use of social media could mitigate the potentially harmful effects of physical distancing, research shows that adolescents perceived social media use during the pandemic as insufficient compensation for meeting friends in real life [54]. A high level of online contact even promoted depressive symptoms [18,24].

### 4.6. Differences According to Gender

In terms of gender differences, female respondents expressed more concerns, which is consistent with the results of many other studies [21,22,24,32,36,38]. Girls were more concerned about the difficulty of returning to normal life and were more likely to express a desire to return to pre-pandemic life. This suggests that female respondents generally felt more burdened by the pandemic measures, which is consistent with other studies [83] and may indicate that female adolescents rely more on face-to-face contact than male adolescents [84]. Female respondents were also more likely to fear failure, which is in line with other studies [68,85]. At the same time, they were more likely to express a desire for a happy life in the future. This seems plausible in light of recent research indicating lower life satisfaction among female respondents [65,68,86]. Female respondents were more likely to report health-related concerns about the future. Similar findings were observed in the recent study by Hawke et al. [71] on health issues and health-related concerns during the COVID-19 pandemic. Moreover, numerous studies have already demonstrated higher levels of psychological distress among female adolescents during the COVID-19 pandemic [39,73,87,88,89,90]. One of the possible reasons for this may be that males are less likely to vocalize their problems to adhere to the traditional male gender role of projecting strength [91].

### 4.7. Differences According to Migration Background

In terms of migration background, pandemic-related concerns were more likely to be expressed by respondents without a migration background. In particular, the wish for the measures to be discontinued stood out. It may be the case that, due to their generally higher socio-economic status [92], young people without a migration background were more involved in social and cultural activities before the pandemic and therefore experienced the restrictions on freedom as more burdensome than young people with a migration background. On the other hand, respondents with a migration background were more likely to express concerns about the future addressing a lack of internal locus of control, particularly fear of failure. There is also evidence in the research literature of a general, pandemic-independent association between migration background and external locus of control [93]. 

### 4.8. Differences According to the Degree of Psychological Distress

Due to the association between pandemic-related concerns and psychological distress in numerous studies [21,24,60,71,86], this aspect was also investigated in our study. Rodríguez-Cano et al. [24] found an association between depressive symptoms and worries about school, financial situation and COVID-19 infection in their study. Similar results were reported by Kim et al. [21]. In our study, psychologically distressed adolescents were also more likely to express school-related concerns. Psychologically distressed adolescents were more likely to report major life goals as hopes for the future, especially life satisfaction. As higher levels of psychological distress are associated with lower quality of life [24,60], it is understandable that psychologically distressed adolescents are more likely to long for a satisfying life.

### 4.9. Implications

The results of this study suggest an increased need for counseling, support and assistance for adolescent school students, and the need to critically examine school-based measures for pandemic containment in the future. The study by Kaltschik et al. [87] demonstrates that even after the end of the COVID-19 pandemic, there is still a need for support among adolescent students, as they showed poorer mental well-being in spring 2022 compared to February 2021. Despite an increase in the demand for help among young people with mental distress [90], young Austrians have shown a reluctance to access professional support services [37]. Therefore, support services for young people should be designed to be easily accessible, with schools being a suitable setting for such support. Services such as the school psychological service and school social work, which are currently only available sporadically or with limited hours [94,95,96], should be expanded and made available in all schools. Outpatient counseling and psychotherapy services, for which young people currently have to wait several months [97], need to be expanded to serve more young people than is currently possible. To make these services accessible to socially disadvantaged young people, an increased number of free counseling hours should be made available at a regional level. As some students benefited from the pandemic-related changes to their daily lives, it is important to consider not only the negative effects of COVID-19 on the mental health and well-being of children and adolescents but also to include the perspective of young people and their families who have benefited from the changes brought about by the lockdown, particularly in the educational domain.

### 4.10. Limitations

This study has several limitations. The first and most important limitation is that the study was conducted based on an online survey. Although the young people were given the opportunity to express their subjective experiences in their own words in the free text boxes, the responses give only a limited picture of the reality of the school students’ lives. The richness of detail that could be generated, for example, by qualitative in-depth interviews, which deepen the narratives and place them in relevant contexts, is not possible with this type of research. However, as a pilot study, our research can provide indications as to which issues could be explored in more depth in a further qualitative interview study.

The online survey format also meant that school students’ responses varied in detail, from very brief keyword responses to more substantial and differentiated responses. It is possible that the willingness to answer could be increased by incentives. In addition, the research questions were very broad and non-specific and did not refer to any time frame. As a result, young people’s responses covered a wide range of life domains and periods, from the near future to the next stages of their lives. Specifying a timeframe when formulating the open-ended questions would narrow down the young people’s responses.

Secondly, our sample ensures a representative selection in terms of gender and migration background. However, other factors, such as age and socioeconomic status, also have an impact on the well-being and mental health of adolescents [36,98]. The subjective socioeconomic status is, for example, associated with subjective well-being, mental health and self-rated health during adolescence [99,100]. Unfortunately, no information on the socioeconomic background was collected in the present study, which should be considered in future studies to avoid bias.

Thirdly, the concerns about and hopes for the future expressed by the young people in our sample cannot be attributed solely to the pandemic, as other factors such as family problems, personal illnesses, etc., may contribute to individual deterioration in mental well-being.

## 5. Conclusions

The findings of our pilot study add to our understanding of what young people are facing at a time of major change and provide insights into their concerns and hopes for the future one year after the beginning of the pandemic. The findings show that young people have concerns about the future in many areas of their lives, which society must address if we are to take young people’s fears and needs seriously. The findings of this pilot study help to raise awareness of the concerns of school students in Austria and can serve as a basis for further in-depth qualitative research.

## Figures and Tables

**Figure 1 healthcare-11-02242-f001:**
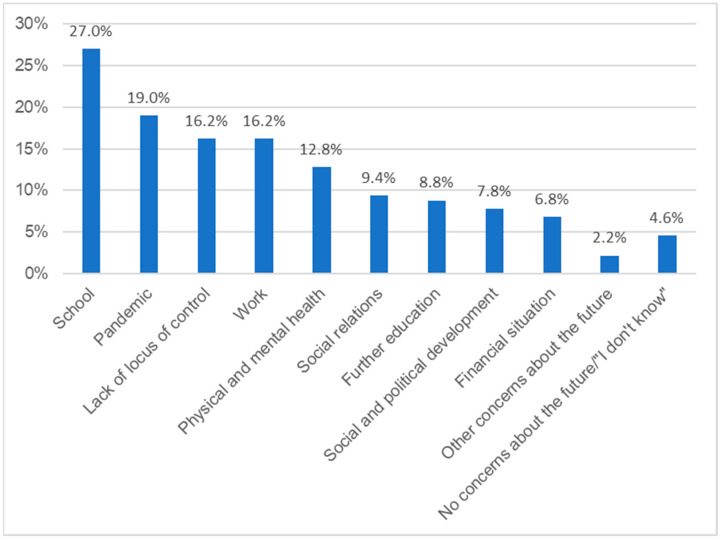
The greatest concerns about the future. The percentages of respondents reporting each main category of concerns about the future that emerged from the data for question 1: “What is your greatest concern when you think about the future?”

**Figure 2 healthcare-11-02242-f002:**
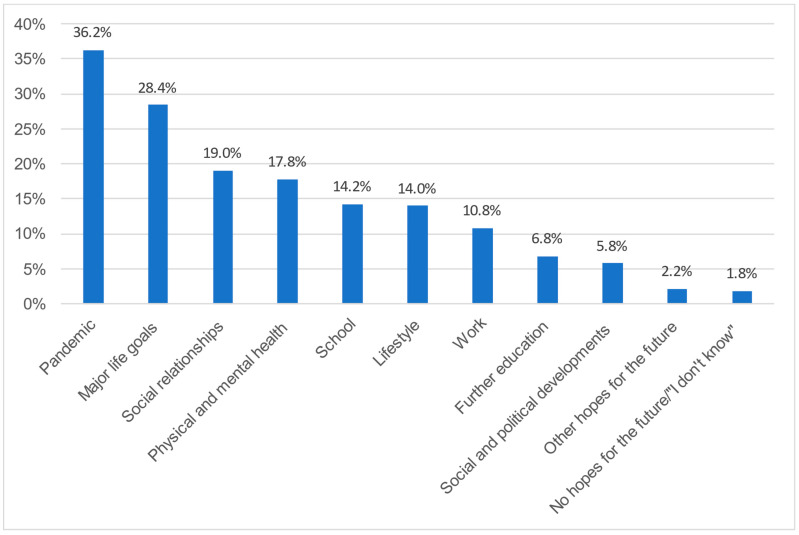
Greatest hopes for the future. The percentages of respondents reporting each main category of hopes for the future that emerged from the data for question 2: “What do you hope for most when you think about the future?”

## Data Availability

The raw data supporting the conclusions of this article will be made available by the authors upon reasonable request after signing a confidentiality agreement.

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
