# Peer review of "Young People and the Future: School Students’ Concerns and Hopes for the Future after One Year of COVID-19 in Austria—Findings of a Mixed-Methods Pilot Study"

_healthcare, 2023, doi:10.3390/healthcare11162242_

Round 1

Reviewer 1 Report

This study “Young People and the Future: School Students’ Concerns and Hopes for the Future after one year of COVID-19 in Austria. Findings of a Mixed-Methods Study“ aimed to gain a deeper insight and understanding of young people’s concerns and hopes for the future through open-ended questions that allowed respondents to express their thoughts freely. Differences in concerns and hopes for the future according to gender, migration background and level of psychological distress were explored. I appreciated author´s work.

This topic seems interesting but, I think the Methods could be understood as a qualitative approach but maybe not as a mixed-methods study.

The results report could also be improved, so there are too many figures to explained this part of information in the manuscript. This excess complicates the understanding of the results. I suggest using a table to summarize all the categories, to clarify the report of the results.

Discussion is too long, I would suggest structuring it in subsections according to the categories presented in Results.

Author Response

Dear reviewer,

thank you very much for taking the time and reviewing our paper. We very much appreciate your valuable feedback. Below we address your comments as implemented in the paper:

  1. With respect to the methods, we believe that a mixed methods approach better describes the applied methods, as we combined qualitative and quantitative data in additional statistical analyses. As the qualitative data were quantified and evaluated for statistical differences, this approach does not represent a purely qualitative method.
  2. We have followed your recommendation to summarise the information from the figures in a table. In addition, we have condensed the results section and reduced the quotations from the original material.
  3. We have also implemented your suggestion to shorten the discussion and make it easier to read by dividing it into sub-sections. Thank you for your suggestion, which makes this central chapter even clearer.

Reviewer 2 Report

Having analysed the paper entitled “Young People and the Future: School Students’ Concerns and Hopes for the Future after one year of COVID-19 in Austria. Findings of a Mixed-Methods Study”, I reached the following conclusions:

1.      The content of the work is consistent with the aim of the work.

2.      The work addresses an important issue related to current health status of adolescents.

3.      Although there are numerous publications devoted to the topic mentioned above, the article manages to fill in the gaps in the already published works, especially with regard to the research method.

4.      It might be wise for the authors to shorten the Introduction section, it is too broad.

5.      The methodology does not raise any objections.

6.      While discussing the results presented in graphs, the authors might want to focus on opinions that were most common/least common. It also might be wise for the authors to consider presenting the part devoted to the qualitative analysis in tables, e.g. in the form of a summary table for each question. This might help to increase the transparency of the discussed section.

7.      The Discussion section was organized in a thorough and correct way.

8.      The Conclusion section should be redrafted. It should not include a discussion of the research idea.

9.      The research literature was correctly selected.

Author Response

Dear reviewer,

thank you for your appreciative words about our work and your much-valued feedback.
Below we briefly discuss the points that have been implemented. We hope that this will enable you to accept our paper.

  1. We have followed your recommendation to shorten the introduction.
  2. We have also implemented your helpful suggestion to present the categories of the qualitative content analysis in a summary We have also condensed the results section and reduced the quotations from the original material.
  3. We have redrafted the conclusion.

Reviewer 3 Report

This study addressed Austrian students’ subjective feelings during the pandemic. The data was collected during the pandemic and was intended as a general survey about students’ psychological well-being at the time. Though the topic is important, the research design and data were limited in its ability to answer the research questions.

First, the research question lacks strong theoretical embedding. Although the authors clearly stated that the study was an exploratory one, lacking theoretical embedding would still makes the reader disoriented as to what is the intended goal of the study. Is their goal to determine the extent to which students were psychologically affected by the pandemic? Or to explore the psychological stressors, so as to develop intervention programs? Or is their goal strictly targeted at “subjective experience”? 

Second, related to the first concern, as was stated in Line 132 by the authors, the study intended to focus on “the subjective experience of adolescents” as contrast to the quantitative research designs adopted by previous studies. So a more appropriate study design seems to be a qualitative one, in which the researchers recruit participants for in-depth interviews and analyze the transcribed texts. However, in the remainder of the paper, the authors still devoted much space to the results of their questionnaire study (quantitative).

Third, the study conducted a questionnaire study, and a group of percentage results were reported and were analyzed by chi-square test. The goal of the questionnaire as well as the quality and suitability of the items were all in doubt. 

Forth, the whole study was more suitably treated as a pilot study, as collecting preliminary reactions to be used as materials in developing more rigorous and more theoretically minded measures. 

Author Response

Dear reviewer,

thank you very much for reviewing our paper and providing valuable comments to improve it. In the following, we would like to briefly address your comments:

1 und 2. The aim of the study was to provide an insight into young people's concerns and hopes for the future as the pandemic progressed. Due to the extraordinary time the data were collected, we did not want to limit the expression of students' concerns and hopes to pre-defined categories. Therefore, a qualitative approach with open-ended questions was chosen. At the same time, we wanted to capture a broad picture of young people's concerns and hopes for the future, representing the views of respondents from different socio-demographic groups. This is why we chose to conduct our study in an online survey format and drew a representative sample by gender and migration background.

We fully agree that qualitative interviews would have been a better approach for an in-depth investigation of subjectively experienced concerns and hopes for the future. However, this would have been feasible only with a very limited number of cases. Our design allowed us to gain deeper insights from a larger sample of Austrian school students that was representative in terms of gender and migrant background. 

In response to your critical comments, we have revised the part of the Introduction that outlines the aim and purpose of our study. Based on your feedback, we decided to present our study as a pilot study. We believe that our findings and experiences from our study will be valuable in informing further in-depth qualitative research, e.g., in designing interview guides (e.g., deciding on the range and content of questions and the intended time frame of the questions) and choosing a sampling design.

We have also added a paragraph in the limitations section to make the drawbacks of the online survey approach more explicit.

  1. The percentages of the categories were formed on the basis of the assigned text passages in the qualitative content analysis. The content analysis was carried out according to the rules of good scientific practice and quality criteria of qualitative social research were taken into account (e.g., validation by the research team, iterative approach to qualitative data analysis).
  2. Thank you for this valuable suggestion, which we have implemented. From our point of view, framing our study as a pilot study with the intention of informing further in-depth qualitative research has given the paper the right angle.

Reviewer 4 Report

This article is more important for understanding both adolescent health and positive psychology. But I still have some questions about the article.

1.  The article may be too long, especially the content about the qualitative results, listing too much, it feels like a report.

2. Table 1 of the article, concerning the region of the study subjects, does not need to list so much, considering that the readers of the article do not know Austria. Using the classification of the study area by the level of economic development might be sufficient.

3. The article mainly considered the impact of migration factors on health but lacked additional socioeconomic variables, such as household income.

Author Response

Dear reviewer,

thank you for your appreciative words about our work and your much-valued feedback.
Below we briefly discuss the points that have been implemented. We hope that this will enable you to accept our paper.

  1. We have shortened the whole paper, especially the results part of the qualitative analyses, as suggested. Results were summarized in Tables and the quotations from the original material were reduced.
  2. We have taken into account your helpful comment that the Austrian regions are too detailed and changed the classification of regions according to major Austrian socio-economic regions.
  3. We agree that the missing information on the socioeconomic status, i.e., the household income, is a major limitation of our study. However, as our study is based solely on adolescents’ self-reports, we decided against asking about household income considering the age and maturity of the surveyed group. Adolescents may have limited knowledge or understanding of household income and financial matters. In future studies asking about alternative indicators of economic status or socio-economic background, such as parents' educational attainment, occupation, or the type of neighborhood the adolescents live in will be considered. We have taken up your critical comment and put more emphasis on this limitation in the revised discussion.

Round 2

Reviewer 3 Report

The authors have revised the manuscript accordingly. Though the study is very preliminary and the contribution is limited, it could still provide some clues for further investigation.  I have no further comments. 

Author Response

Thank you for reviewing our article a second time and for your positive assessment.

Reviewer 4 Report

The new version is much better. This article has some value for adolescent health and positive psychology. The research methodology is also innovative to some extent. I would still recommend a shorter and more precise version of the article before publication.

Author Response

Thank you for reviewing our paper a second time and for your positive assessment.

As you recommended, we have shortened the whole paper again, especially the part about the qualitative results of the study. We hope you can now endorse publication.
